# 2024 ‘Key Reflections’ on Sadi Carnot’s 1824 ‘*Réflexions*’ and 200 Year Legacy

**DOI:** 10.3390/e27050502

**Published:** 2025-05-07

**Authors:** Milivoje M. Kostic

**Affiliations:** Department of Mechanical Engineering, Northern Illinois University, DeKalb, IL 60115, USA; kostic@niu.edu

**Keywords:** Sadi Carnot, Carnot cycle, reversibility, heat engine, contradiction impossibility, maximum engine efficiency, thermodynamics, second law of thermodynamics

## Abstract

This author is not a philosopher nor a historian of science, but an engineering thermodynamicist. In that regard, and in addition to various philosophical “*why and how*” treatises and existing historical analyses, the physical and logical “*what it is*” *reflections, as sequential Key Points*, where *a key Sadi Carnot reasoning infers the next one*, along with novel contributions and original generalizations, are presented. We need to keep in mind that in Sadi Carnot’s time (early 1800s), steam engines were inefficient (below 5%, so the heat in and out was comparable within experimental uncertainty, as if *caloric* were conserved), the conservation of *caloric* flourished (might be a fortunate misconception leading to the critical analogy with the waterwheel), and many critical thermal concepts, including the conservation of energy (The *First Law*), were not even established. *If Clausius and Kelvin earned the title* “*Fathers of thermodynamics*”, *then Sadi Carnot was ‘the ingenious’* “*Forefather of thermodynamics-to-become*”.

## 1. Introduction




*“The motive power of heat is independent of the agents employed to realize it; its quantity is fired solely by the temperatures of the bodies between which is effected, finally, the transfer of the caloric.”—by Sadi Carnot, 1824*
(English Translation by Robert H. Thurston [1]).

***

In 2024, the thermodynamic community celebrated *Sadi Carnot’s Legacy* and the *200th Anniversary* of his only and famous ‘*Réflexions*’ publications, which was not appreciated at his time, and may only be truly comprehended by a few, even nowadays [1,2,3,4,5]. It should be noted that original and ingenious Sady Carnot’s discoveries were expounded and formalized decades later by Kelvin [6] and mostly by Clausius [7], while some persistent post-misconceptions and fallacies by others are also presented here as ‘*Miss Points*’. Even the ‘*Carnot Cycle efficiency’ per se* is fundamentally misplaced as reasoned below.

Here, we present this author’s contribution with ‘2024 *Key Reflections*’ and the plenary presentation of his lifelong work at *Sadi Carnot’s Legacy International Colloquium* at École Polytechnique in France [3,8].

As Anthony Legget, a Nobel laureate, commented, “*Mathematical convenience versus physical insight […] that theorists are far too fond of fancy formalisms which are mathematically streamlined but whose connection with physics is at best at several removes […] heartfully agreed with Philippe Nozieres that ‘only simple qualitative arguments can reveal the fundamental physics’*”. In that regard, instead of an extensive review of literature or an ‘extravagant’ historical or philosophical formal methodology (as performed by many), here, mostly thermodynamic, specific, and simple reasoning, but with deep intuitive comprehension, were emphasized by a lifelong engineering thermodynamicist, for the first time as such in the literature.

It is well known that “*Thermodynamics is not easy to understand ‘the first time around’*”. Thermodynamics is confusing until it is comprehended, and this work is no exception. Sometimes, interpretation by different experts may result in further confusion. Namely, not all scientists/physicists and engineers are thermodynamicists, and a “*thermodynamic historian*” is not a “thermodynamicist”, but a “historian”, often with elegant style and abstract methods, but sometimes without due comprehension of tacit thermodynamic fundamentals. We are often trapped in our own thoughts and words (especially in emerging new concepts and if non-native) and subtle holistic meanings are to be read “between the lines”. Sometimes, highly accomplished scientists in their fields do not fully comprehend the essence of thermodynamics, especially if related to the *Second Law* and *entropy*.

The objective of this article is not to formally review Carnot’s ‘*Réflexions*’, nor the vast amount of related literature by others (which was already, in part, carried out by this author [9,10,11] and many others, for example [12,13,14,15,16]), but to present this author’s long-contemplated reflections, with critical thermodynamic insights and logical reasonings, and to put “key thermodynamic concepts” into a contemporary perspective. Namely, selective, physical, and sequential ‘*Key Points*’, as judged by this author, where the ingenious Sadi Carnot’s reasoning infers the next one, along with ‘*Miss Points*’, presenting persistent post-misconceptions and fallacies by others (that needed to be underscored and ‘*put to rest*’), are presented as such for the first time. The emphasis is on engineering and phenomenological thermodynamics (on fundamental substance instead of formal methodology and style), and not on a philosophical and historical review and ‘extension’ of Carnot’s work (already carried out by many). Therefore, only selected and related publications by this author are referenced.

Sadi Carnot, at age 28 in 1824, published the now famous “*Réflexions sur la puissance motrice du feu*” (English translation, “*Reflections on the Motive Power of Fire* [1]”). His ingenious reasoning of reversible processes and cycles, and maximum “*heat-to-power efficiency*” laid the foundations for the *Second Law of thermodynamics*, before *The First Law* of energy conservation was even formulated (in the 1840s), and long before thermodynamic concepts were established [6,7] (in the 1850s and later elsewhere). Sadi Carnot, who died in 1832 at age 36 during a cholera epidemic, could not had been aware of the immense implications of his ingenious reasoning at that time. No wonder Sadi Carnot’s masterpiece, regardless of its flawed assumption of *conservation of caloric*, was not appreciated at his time, when his ingenious reasoning of ideal “*heat engine reversible cycles*” was not fully recognized; it may only be truly comprehended by a few, even nowadays.

Before this author’s “*2024 ‘Key Reflections’ on the 1824 Sadi Carnot’s ‘Réflexions’*” are presented here, a brief introduction, based on his prior publications [9,10,11], is given next to revisit the essential concepts. Note that Carnot’s *Réflexions* focused on ideal, reversible cycles and maximum possible efficiency, hence the reversible processes and cycles are assumed here, unless specifically stated otherwise (as dissipative or irreversible). Furthermore, the concepts of *heat* and *work* were not well formalized in Carnot’s time, but *caloric* and *motive power* were used instead; the latter, as work rate or in short ‘*work’*, is often used here (e.g., heat or work for duration of a cycle or for some time period, should be self-evident).

Sadi Carnot gave a full and accurate reasoning of heat engine cyclic processes and their limitations of “*converting heat to* [work] *motive-power*” at a time when caloric theory was flourishing and almost two decades before equivalency between work and heat was experimentally established by Joule and others, in the 1840s (see elsewhere).

At that time, when the energy conservation law was not known, heat was considered an indestructible caloric, and heat engines were in their initial stage of development with an efficiency of less than 5%, confusion and speculations flourished. Can efficiency be improved by different temperatures or pressures, a different working substance than water, or some different mode of operation other than pistons and cylinders? With ingenious and far-reaching reasoning, Sadi Carnot answered all of those questions and logically reasoned (thus proved) that the maximum, limiting efficiency of heat engines does not depend on the medium used in the engine or its design, but only depends on (and increases with) the temperature difference between the heat source and cooling medium or heat sink (however, not linearly), similarly to the waterwheel work–power dependence on the waterfall height difference at a given water flow rate, see Equation (1) and Figure 1 (explicit formulas were developed after Carnot’s followers’ work [6,7] and elsewhere; see also *Miss Point 1* and *Key NOVEL-Point 4*).

Most importantly, Carnot introduced reversible processes and cycles and, with the ingenious reasoning of “*Contradiction impossibility*”, see *Key NOVEL-Point 3,* proved that the maximum heat engine efficiency is achieved by any reversible cycle, thus, all must have the same maximum possible efficiency ([1,9,10], see also *Key NOVEL-Point* 2), i.e.,

“*The motive power of heat is independent of the agents employed to realize it; its quantity is fired solely by the temperatures of the bodies between which is effected, finally, the transfer of the caloric*” [1]. Namely,(1)$$W_{C}=W_{netOUT}=Q_{IN}\cdot f_{C}(T_{H},T_{L});\;\;\;\;\eta_{C}={\left.\frac{W_{netOUT}}{Q_{IN}}\right|}_{Max}={\left.\underset{\mathrm{Qualitative}\;\mathrm{function}\;}{\underset{︸}{f_{C}(T_{H},T_{L})}}\right|}_{\mathrm{Rev}.}$$

The Carnot cycle consists of four reversible processes (see Figure 2): isothermal heating and expansion at a constant *high* temperature *T_H_* (also referring to ‘*hot* reservoir’ elsewhere; *process* 1–2); adiabatic expansion to achieve a *low* temperature *T_L_* (also *T_L_* ≡ *T_C_* referring to ‘*cold* reservoir’ elsewhere; *process* 2–3); isothermal cooling and compression at a constant low temperature *T_L_* (*process* 3–4); and adiabatic compression to achieve a high temperature *T_H_* and complete the cycle (*process* 4–1).

All processes are reversible; thus, the cycle could be reversed, without additional external intervention, along the same path and with the same quantities of all the heats and works in opposite directions (*in-to-out* and vice versa), see Equation (2) and Figure 3, i.e.:(2)$$\left\{Q_{H},Q_{L},W_{C}\right\}\underset{\mathrm{IF}\;\mathrm{REVERESED}}{\underset{︸}{\Leftrightarrow }}\left\{-Q_{H},-Q_{L},-W_{C}\right\}$$

The concept and consequences of a process and cycle reversibility are the most ingenious and far-reaching, see [5,9,10] (see also *Key NOVEL-Point 1*). Sadi Carnot’s simple and logical reasoning that mechanical work is extracted in a heat engine due to the heat passing from a high to low temperature (see also *Key Point I*) led him to a very logical conclusion that any heat transfer from a high to low temperature (like in a heat exchanger) without extracting the possible work (like in a reversible heat engine) will be a waste of work potential, so he deduced that any heat transfer in an ideal, perfect heat engine must be at an infinitesimally small temperature difference (no loss of caloric fall potential), achieved via mechanical compression or the expansion of the working medium (required temperature adjustment without heat transfer), as Carnot ingeniously advised in full details [1] (see also *Key Point II*).

Then, Sadi Carnot expended his logical reasoning to conclude that all reversible (ideal) heat engines must have an equal and maximum possible efficiency, otherwise, if reversed, the impossible “*creation of conserved quantities*” would be achieved, see all details in [1,9,10] and elsewhere (see also *Key Point V* and *Key NOVEL-Point 2*). What ‘a simple’ and logical ingenious reasoning!

Carnot’s reasoning proves that a reversible cycle cannot have a lower efficiency (power output relative to heat input) than any other cycle, thus all reversible cycles must have the same maximum possible power–efficiency for the given temperatures of the two thermal reservoirs, independently from anything else, including the nature of the heat engine design and its agent undergoing the cyclic process (see all relevant specifics in [1,9,10] and elsewhere; see also *Key Point V* and *Key NOVEL-Point 2*).

Since the irreversible cycles could not be reversed, they may (and do) have lower than maximum reversible efficiency up to zero (no net work produced, if all work potential is dissipated to heat) or even negative (external work input required to run such a “parasite” engine that will dissipate such work into heat, in addition to the original work potential dissipation), i.e.:(3)$$\eta_{Irr}<\underset{\mathrm{Reversible}}{\underset{︸}{\eta_{\mathrm{Rev}}=\eta_{\max}=f_{C}(T_{H},T_{L})}}$$

Carnot did not provide a quantitative, but a qualitative correlation for the ideal heat engine power–efficiency. Note that this combined, empirical efficiency (“net work output per heat input”) includes not only heat engine but also the heat reservoirs, and in that regard is misleading (see *Key NOVEL-Point* 4). Sadi Carnot accurately specified all conditions that must be satisfied to achieve reversibility and the maximum efficiency: the need for “*re-establishing temperature equilibrium for caloric transfer*”, i.e., reversible processes, where the reversible heat transfer has to be achieved at a negligibly small (in the limit of zero) temperature difference at both temperature levels, at *T_H_*, high temperature for heat source (reversible heating), and at *T_L_*, low temperature for heat sink (reversible cooling of heat engine medium), see Figure 2 and Figure 3, otherwise the work potential during the heat transfer will be irreversibly lost due to the temperature difference (the main Carnot *cause-and-effect* reasoning), see also *Key Point II*.

Sadi Carnot reasoned that mechanical expansion and compression are needed to decrease and increase the temperature of the engine medium to match the low and high temperature of the thermal reservoirs, respectively, and thus provide for the reversible heat transfer [1].

Carnot then reasoned that in limiting cases, such as an ideal cycles, it could be reversed using the prior obtained work, to transfer back the caloric (heat) from low- to high-temperature thermal reservoirs, thus laying the foundations for the refrigeration cycles (cooling and refrigeration/heat pumps) as ‘reversed’ heat engine cycles, see Figure 3 and Equation (2). 

Sadi Carnot’s reasoning of “*heat engine reversible cycles and their maximum efficiency*” is importance-wise on par with Einstein’s *Relativity theory* in modern times, see Equation (4). It may be among the most important correlations in natural sciences that led to the discovery of *entropy* and the *Second Law* of thermodynamics, among others. This claim was stated by this author, ‘symbolically’ expressed by Equation (4), and named as the ‘*Carnot* (ratio) *Equality’* in prior publications [9,11] to emphasize the invaluable but not well-recognized importance of the *Q/T* = *constant* correlation, renamed here as ‘*Carnot-Clausius Equality’*, as the precursor to and to resemble the ‘*Clausius Equality*’ cycle integral, both formalized by Clausius [7] based on Carnot’s *Réflexions* [1].(4)$$\left.\left\{\begin{array}{c} by\;Carnot\prime s\;followers\;[Clausius\;(1854)]\\ \underset{CarnotEquality(CtEq)}{\underbrace{\overbrace{{\frac{Q(T)}{Q(T_{0})}=\frac{f(T)}{f(T_{0})}|}_{f(T)=T}=\frac{T}{T_{0}}=\frac{Q}{Q_{0}}}}}\\ \underbrace{or\;\frac{Q}{T}=\frac{Q_{0}}{T_{0}}i.e.,\frac{Q}{T}=constant} \end{array}\right.\right\}\begin{array}{cc} \mathrm{Essentially} & Einstein\\ \mathrm{Important}\;\mathrm{as} & \left(1905\right)\\ \overbrace{<?>} & \overbrace{\left\{mc^{2}\right\}} \end{array}$$

The ‘*Key Reflections*’ presented next are founded on Sadi Carnot’s ‘*Réflexions*’ (English translation by Robert H. Thurston [1] and elsewhere) and on the developments of thermodynamics by the pioneers [6,7] and others, emphasizing this author’s views as a lifelong engineering thermodynamicist [5,8,9,10,11] as a complement to existing science historians and science philosophers’ analyses, e.g., [12,13,14,15,16].

Therefore, a key thermodynamic logic is used to recognize and infer the most probable sequential developments of Sadi Carnot’s ingenious discoveries, as well as to reflect on the related analyses and misconceptions by others, considering the current state of knowledge—since now, we have the advantage to look at the historical developments more comprehensively and objectively than the pioneers. The sequential ‘*Key Points*’, where key reasoning by Sadi Carnot infers the next one, along with ‘*Miss Points*’ (persistent post-misconceptions and fallacies by others), including novel contributions and original generalizations by this author, as ‘*Key NOVEL-Points*’ with ‘*Key Takeaways*’*,* are presented next.

## 2. *Key Points*: Most Probable Sequential Developments of Sadi Carnot’s Ingenious Discoveries

I.*The source of the heat engine “motive power” is “caloric fall” (“temperature fall” or temperature difference in the caloric)*;II.*The “temperature fall,” as a source of the engine motive power, should “not be wasted,” but minimized in any “workless heat-transfer process”*;III.*All motive frictions and other dissipative processes should be minimized in order to maximize the engine power and efficiency*;IV.*Reversible Cycles: Isothermal heat transfer and other frictionless processes make an engine process or cycle reversible*;V.*Reversible cycles must all have equal and maximum efficiency*.

***Key Point I:*** *The source of the heat engine “motive power” is “caloric fall” (“temperature fall” or temperature difference in the caloric).*

Hot *caloric* (*heat* at high temperature) is the cause and source of the *motive power* (produced work) of the steam engines (the heat engines in general). Since at the time, the *caloric* was believed to be conserved (might have been a fortunate misconception leading to the critical analogy with the waterwheel, see Figure 1), Sadi Carnot inferred that its hotness was producing the motive power while being cooled: “*motive power due to the [temperature] ‘fall of the caloric’*”.
***KEYNOTE* 1:** If the temperature of the heat is higher than the surroundings, the higher ‘temperature fall’ through an engine, the higher the motive power will be (the more power per unit of heat flow, the more efficient engine), analogous to the higher power from more “elevation-fall” from a higher elevation through a waterwheel per unit of water flow. In Sadi Carnot’s words, “*The temperature of the fluid should be made as high as possible, in order to obtain a great fall of caloric, and consequently a large production of motive power* [1]”.

***Key Point II:**** The “temperature fall,” as a source of engine motive power, should “not be wasted”, but minimized in any “workless heat transfer process”*.

This is the most critical and ingenious reasoning by Sadi Carnot, “*wherever there exists a difference of temperature, motive power can be produced*” [1], which led to the inference of ideal reversible cycles, the most critical concept of Carnot’s discovery. If the temperature difference is the cause and source of the motive power, then, if it is “consumed” during the heat transfer without work extraction, then its work potential will be lost, so the temperature difference during the heat transfer should be minimized, i.e., be infinitesimal; ideally the heat transfer should be isothermal: “*need for reestablishing temperature equilibrium for caloric transfer … in the bodies employed to realize the motive power of heat there should not occur any change of temperature which may not be due to a change of volume* [1]”.

***Key Point III:**** All motive frictions and other dissipative processes should be minimized in order to maximize the engine power and efficiency. The mechanical processes should ideally be frictionless to avoid work waste, i.e., dissipation losses*.

***Key Point IV:**** Reversible Cycles: Isothermal heat transfer and other frictionless processes make an engine process or cycle reversible*.

This is a monumental and crucial “*reversibility concept*” with far-reaching consequences. Reversible processes and cycles could effortlessly (without additional external compensation) reverse back-and-forth in perpetuity (like *perpetual motion*), therefore without any degradation or loss, being perfect and with maximum possible 100% efficiency. They take place at infinitesimal potential differences in either direction (in limiting the equipotential process, no potential loss of quality) and without any quantity nor quality degradation, including conservation of the motive power or work potential. More details can be found in *Key NOVEL-Point 1* and elsewhere.

***Key Point V:**** Reversible cycles must all have an equal and maximum efficiency*.

This “*key discovery*” (*Carnot Theorem*) was ingeniously inferred by Sadi Carnot by logical reasoning that otherwise would result in the creation of [assumed] conserved caloric and/or perpetual motion: “*the maximum of motive power resulting from the employment of steam is also the maximum of motive power realizable by any means whatever* [1]”. This powerful insight is the most important and ingenious reasoning of Sadi Carnot, and it has far-reaching consequences, as demonstrated much later by Kelvin [6] and Clausius [7], and other followers of Sadi Carnot. This is further elucidated in *Key NOVEL-Point* 2, and in *Key NOVEL-Point* 3, it is further generalized as “*Reversible Contradiction impossibility,*” see also Figure 4.

The selected and persistent post-misconceptions and fallacies by others are presented as ‘*Miss Points*’:

## 3. *Miss Points*: Persistent Post-Misconceptions and Fallacies by Others

*The well-known Carnot efficiency formula,*  
${\eta_{Carnot}=W}_{Rev|Max}/Q_{H}=\left(1-T_{L}/T_{H}\right)$*, was not established by Sadi Carnot, but much later by Kelvin and Clausius;*
*The cause and source for motive power is the temperature difference, in principle, but is not linearly dependent, as mis-stated by some;*

*The heat transferred out of the Carnot cycle at a lower temperature is “not a waste heat” as often stated, but it is a “useful quantity”, necessary for the completion of the cycle;*

*Sadi Carnot could not have been thinking of “any other caloric” but heat to imply “entropy-like quantity”, as speculated by some.*


***Miss Point 1:*** The well-known Carnot efficiency formula, ${\eta_{Carnot}=W}_{Rev|Max}/Q_{H}=\left(1-T_{L}/T_{H}\right)$, was not established by Sadi Carnot, but much later by Kelvin and Clausius.

In 1824, Sadi Carnot inferred the maximum heat engine power efficiency as an implicit function of thermal source and sink reservoirs’ *high* and *low, t_H_* and *t_L_*, temperatures only [${\eta_{Rev|Max}=W}_{Rev|Max}/Q_{H}=f_{C}\left(t_{H},t_{L}\right)$]. Note that the absolute temperature concept was not known at that time. However, the well-known *Carnot efficiency* formula, ${\eta_{Carnot}=W}_{Rev|Max}/Q_{H}=\left(1-T_{L}/T_{H}\right)$, with absolute temperature, sometimes attributed as having been developed by Sadi Carnot, was actually developed much later, in the 1850s, first by Kelvin [6] using ideal gas, and later by Clausius [7] in general, and named “*Carnot efficiency*”.

Paradoxically, it is shown here that Carnot, Kelvin, and Clausius’ concepts of the maximum reversible cycle efficiency are misplaced, since fundamentally, the Carnot cycle efficiency is not the “reversible cycle efficiency” per se, but a power-per-heat ‘*coefficient of performance*, *COP* < *1’*, that includes both the heat engine and thermal reservoir efficiencies, to be decoupled, see *Key NOVEL-Point* 4. It is fundamentally like its inverse, the heat-per-power *COP > 1* of the heat pump. Essentially, it is a thermal energy source *‘work-potential efficiency’*—see more details in *Key NOVEL-Point* 4. After all, the Carnot cycle efficiency does not depend on the cycle in any way, but on the thermal reservoirs’ temperatures only.

***Miss Point 2:**** The cause and source for the motive power is the temperature difference, in principle, but is not linearly dependent, as misattributed by some*.

Carnot stated that the temperature difference is, in principle, the cause and source for the motive power, but not directly, linearly dependent, as misquoted by some [$W_{Max}=Q_{H}\cdot f_{C}\left(t_{H},t_{L}\right)\;\neq \;f(t_{H}-t_{L}$), i.e., not the function of Δ*t* = $\;t_{H}-t_{L}\;only$]. As stated by Sadi Carnot, “*In the fall of caloric the motive power undoubtedly increases with the difference of temperature between the warm and the cold bodies; but we do not know whether it is proportional to this difference. … The fall of caloric produces more motive power at inferior than at superior temperatures*” [1].

***Miss Point 3:**** The heat transferred out of the Carnot cycle at a lower temperature is “not a waste heat” as often stated, but it is a “useful quantity”, necessary for the completion of the cycle*.

The heat transferred out of the ideal Carnot cycle at a lower temperature is “*not a waste*” as often stated, but necessary for the completion of the cycle (the entropy balance), and therefore necessary and a *useful quantity*. As stated by Sadi Carnot, “*… without ‘the cold’ the heat would be useless* [1] (see also elswhere)”. The only waste is the additional heat generated via irreversible work dissipation, accompanied by *entropy generation* in real cycles, which must also be taken out to complete the cycle.

***Miss Point 4:**** Sadi Carnot could not have been thinking of “any ‘other’ caloric” but heat to imply the “entropy-like quantity”, as speculated by some*.

We need to keep in mind that in Sadi Carnot’s time (the early 1800s), the steam engines were inefficient (below 5%, so the heat in and heat out were comparable within experimental uncertainty, as if *caloric* is conserved), the *conservation of caloric* flourished, and many critical thermal concepts, including the conservation of energy (*The First Law*), were not even established. At that time, the *entropy* concept was not even remotely known. Therefore, Sadi Carnot could not have been thinking of “any ‘*other’* caloric” but heat to imply the “entropy-like quantity”, as speculated by some—see *Key NOVEL-Point 6* for more details.

Novel contributions with deeper physical insights and related generalization by this author are formalized in the following ‘*Key NOVEL-Points*’.

## 4. *Key NOVEL-Points*: Novel Contributions and Original Generalizations

*“Reversible and Reverse” Processes and Cycles Dissected*;*Maximum Efficiency and “Reversible Equivalency” Scrutinized*;*Reversible Contradiction Impossibility (“Reductio ad absurdum”)*;*Reversible Carnot Cycle Efficiency Is Misplaced—It is NOT the “Cycle efficiency” ‘*per se*’, but a “Thermal energy-source ‘work-potential efficiency’”*;*The Carnot–Clausius [Ratio] Equality (CCE) and Clausius Equality (Cyclic integral) are special cases of relevant “Entropy boundary integral” for reversible stationary processes*;*The ‘caloric’ is transformable to work and cannot be ‘extended and renamed as entropy’, which is ‘the final transformation’*.

***Key NOVEL-Point 1:**** “Reversible and Reverse” Processes and Cycles Dissected*.

Ideal and perfect “*Reversible processes*” take place at infinitesimal potential differences (temperature, pressure, and similar) at any instant within and between a system and its boundary surroundings, but they may and do change in time (*process* is a change in time). Namely, the spatial gradients are virtually zero at any instant, while the time gradients and related fluxes may be arbitrary, as they are driven by ideal boundary surroundings and facilitated by ideal arbitrary (or infinite) transport coefficients. Therefore, the potential qualities of the flux quantities (heat and different kinds of works) are not degraded, but equipotentially transferred and stored between the system and its boundary surroundings, and thereby ‘truly’ conserved in every way. However, in time, due to the unavoidable irreversible dissipation of work to the generated heat, accompanied by a generation of entropy, all real processes between the interacting systems (including the relevant surroundings) are asymptotically approaching the common equilibrium with zero mutual work potential and maximum mutual entropy.

Namely, if an elastic, ideal gas, or ideal spring is *reversibly* compressed, then the pressure may change in time, but is equal everywhere at any instant across the system and the boundary surroundings (equipotential driving force at any instant). Similarly, if the heat is *reversibly* transferred, the temperature may change in time, but it is equal everywhere within the system at any instant, and if it varies in time, it is driven by the varying, but spatially equipotential surrounding temperature, so that the energy potential quality is stored and conserved everywhere in every way (it may be reversed *back-and-forth in perpetuity* without additional external compensation).

Note that the ‘time and energy-rates’ are irrelevant, per se, for the reversible analysis of energy balances and properties between the initial and final states, being independent of the process type and path of how the final state is achieved, either reversibly or irreversibly, the former being simpler and more suitable for analysis than the latter.

A *Cycle* is a special case of a *quasi-stationary process* when the flow inlet and outlet quantities are the same (feed into each other) and close the cycle. Like a stationary process, a cycle does not accumulate flux quantities and may repeat and last in perpetuity (quasi-stationary). Note that all processes, particle-wise, are transient in time (in the *Lagrangian* sense, from the inlet to the outlet), but for the steady-state or stationary processes (in the *Eulerian* sense), the properties do not change in time (zero temporal gradients) at a fixed location, and for a cyclic process, the flow inlet’s quantities are the same as the outlet’s (since they feed into each other).

A “*Reverse*” concept is independent of and should not be confused with the *reversible* concept. If reversible, a *reverse process* could be reversed using prior, related process work (with an infinitesimal change in the potential difference in opposite directions, and without additional external work compensation), while to reverse an *irreversible* process, it would require additional, external work compensation.

A “*Reverse process*” and/or “*Reverse cycle*” would take place if the driving (forced) potentials of a reversible process or a cycle were reversed (by an infinitesimal change in opposite directions), then such a reversible process would be reversed with all the quantities changing direction from the input to output (and vice versa, e.g., a *refrigeration cycle* is a “*reverse*” of a *power cycle* or vice versa, see Equation (2) and Figure 3). For stationary processes, there are no temporal gradients or accumulation of flux quantities within a system, and for quasi-stationary cycles, there is no accumulation of flux quantities after the completion of a cycle. The input and output quantities would be conserved and could be *reversed back-and-fort in perpetuity* like a perpetual motion.

In reality, there is a need for at least an infinitesimal temperature difference (and/or pressure and similar) to provide a *process ‘sense of direction’*, and to resolve directional ambiguity by chance. Therefore, every process must be at least infinitesimally irreversible (infinitesimally imperfect), with the *reversibility* being an asymptotic, limiting ideal concept. For this reason, even a reversible equilibrium is unachievable; like an absolute zero temperature or any other ideal concept, it can only be approached asymptotically.

***Key NOVEL-Point 2:**** Maximum Efficiency and “Reversible Equivalency” Scrutinized*.

The maximum efficiency of an energy process or cycle entails the *maximum possible work extraction* from a system while coming to an equilibrium with a reference system, usually the surroundings; or the *minimum possible work expenditure* in a *reverse process* of the formation of original system (from within the same reference state), see Figure 4—**Center**. Since the reversible processes do not degrade any potential quality and could be reversed without external intervention, the two works must be the same, with opposite signs only, for all reversible processes, and they represent the maximum *work potential* (WP) for the given non-equilibrium conditions. Therefore, the reversible processes are perfect, and *equally and maximally 100% efficient*. They define the concept of “*Reversible Equivalency*”—the *‘true quantity and quality equality’* of input and output, where relevant quantities and qualities are conserved in perpetuity. In real processes, there will be some work dissipation losses (degradation of work with its dissipative conversion to heat), so that *less work* would be extracted than the maximum possible, and *more work* would be needed than the minimum required, thus reducing the maximum possible efficiency for real, *irreversible processes* (see Equation (3) and Figure 4—**Left**). The reversible processes take place in virtual equipotential conditions (virtually the same temperature, pressure, etc.; they are equipotential locally at any time, thus being reversible at any time), but they may vary in time with time-variable systems and surrounding properties. Therefore, the ‘potential quality’ of all relevant quantities would be equipotentially transferred and stored at any time, i.e., conserved without any degradation (without any dissipation) and could be effortlessly reversed back-and-forth (without additional external compensation).

With the infinitesimal reversal of the relevant potentials, all flux quantities will change directions while conserving the quantities and qualities. Therefore, the work extraction in a *reversible cycle (i.e., work potential*, WP) would be equal to the expenditure or formation work in its ‘*reverse*’ *reversible cycle* (see Equation (2) and Figure 3 and Figure 4). Furthermore, all reversible cycles must have an equal and maximum 100% efficiency, otherwise any *‘under-achieving’* reversible cycle (with lower work extraction than another [reference] cycle), when reversed, would consume less work than the reference cycle, and thus be *‘over-achieving’* with a higher-than-reference 100% maximum efficiency, resulting in a ‘*Contradiction impossibility’*, see *Key NOVEL-Point* 3 next.

***Key NOVEL-Point 3*:*** Reversible Contradiction Impossibility (“Reductio ad absurdum”)*.

As stated above, the reversible efficiency implies the *maximum* work extraction and *minimum* work expenditure in the reverse process; therefore, the two must be equivalent for the given conditions, thus establishing the *Reversible Equivalency*, see also *Key NOVEL-Point 2*. Otherwise, any reversible cycle “*under-achievement*” (obtaining less than the maximum possible) would become an “*over-achievement*” when such a cycle is reversibly ‘reversed’ (accomplishing with less than the minimum required), and such a “*reverse over-achievement*” would be physically impossible—it would be the “*Reversible Equivalency* violation” and may violate the *conservation laws*, thus implying the “*Reversible Contradiction Impossibility*” of a well-known fact, see Figure 4—right (impossible *‘miracle* GAIN’).

Namely, the “*Reversible Contradiction Impossibility*” (an *under-achieving* reversible process when reversed) would become an *impossible over-achieving* reversible process and could result in numerous consequences. Namely, the miraculous creation of ‘perpetual motion’ or the creation of an assumed ‘conserved caloric’ (regardless of Sadi Carnot’s misconception of *caloric* conservation) and other impossible processes, like spontaneous heat transfer from lower to higher temperatures, etc. The ‘Reversible Contradiction Impossibility’ is so strong and universal a concept that any pertinent or quasi-relevant criteria, even if misunderstood, like the conservation of caloric, will be sufficient to reason fundamental inferences [1].

Further consequences of the “*Reversible Contradiction Impossibility*” would be the spontaneous generation of a conserved quantity, or the generation of non-equilibrium work potential, or energy transfer from a lower to higher potential, like spontaneous heat transfer from a lower to higher temperature and the generation of a thermal non-equilibrium, i.e., the impossible destruction of entropy, see more details in [9,10,11] and elsewhere. Or, in general, the spontaneous creation of a non-equilibrium from within an equilibrium being the physical contradiction of the always observed “spontaneous process, forcing-direction from non-equilibrium towards mutual equilibrium,” and never experienced otherwise. It will amount to the “*forced*-*directionality contradiction*” of the *irreversible process directionality* from a higher to lower potential towards mutual equilibrium, as well as the *impossibility to reverse dissipation*.
***KEYNOTE* 2:** It would be logically and otherwise impossible and absurd (“*Reductio ad absurdum*”) to have a spontaneous process “*the one way and/or the opposite way*” arbitrarily in opposite directions, by chance (i.e., to have heat transfer “*from hot-to-cold* or *from-cold-to-hot*” or “*forcing in one direction and accelerating in opposite direction*,” by chance). *It would be a violation of the Second Law of thermodynamics (2LT).*

The reversible processes are equipotential and therefore do not degrade a non-equilibrium, but store and/or convert one kind to another, like a reversible cycle converts ‘*heat at high temperature*’ to ‘*work plus heat at lower temperature*’, and in reverse in perpetuity, defined in [11] as the “*Carnot-Clausius Heat-Work Equivalency, CCHWE*” (‘potential-like’ heat at a high temperature is equivalent and converts to ‘kinetic-like’ work plus heat at lower temperature, and vice versa—analogous to a reversible pendulum, converting potential to kinetic energy, and in reverse, in perpetuity), see Figure 5. The *CCHWE* is a fundamental and autonomous physical concept, independent of any process or device [11].

Therefore, all reversible processes are perfectly “equivalent in every way” and the most efficient, without any dissipative degradation.

***Key NOVEL-Point 4*:*** Reversible Carnot Cycle Efficiency Is Misplaced—It is NOT the “Cycle efficiency” ‘per se’, but a “Thermal energy-source ‘work-potential efficiency’”*.

The reversible processes and cycles, as a matter of concept, are 100% perfect without any degradation and must be equally and perfectly (maximally) efficient, not over nor below 100% efficiency (would be the *Reversible Contradiction Impossibility*). Therefore, all reversible processes and cycles have 100% “true quantity and quality” efficiency—they extract 100% of the “available work potential”, as does any ideal waterwheel and any other reversible energy transformer (e.g., engine or motor). The 100% perfect “true reversible efficiency” [11 (*CCHWE*)], see Figure 6—**Right**, should not be confused with the “maximum work-thermal efficiency” of a thermal energy source, which represents the “work potential of heat” or Exergy of heat (or non-equilibrium thermal energy) of the relevant thermal reservoirs [*E_x_* = *W_Rev_*_|*Max*_ = *Q*(1 − *T*_0_/*T_H_*), where *T_C_* = *T*_0_], see Figure 6—**Left**, as their property-like quantity, being independent of any heat engine or energy device.

Sadi Carnot [1] and his followers, including Kelvin [6] and Clausius [7], ironically referred to the maximum heat engine cycle efficiency (that they “agonizingly” developed at the time, when most thermal concepts were unknown) with the *absurd conclusion* that “*it does not depend on the cycle design itself nor its operation mode* [in any way whatsoever]”, hence, it is not related to the cycle in any way, which is the proof that it is not the efficiency of the ideal Carnot cycle *per se*. Therefore, their attribution is misplaced, since the efficiency they developed *should have referred to* the “maximum motive power or *‘work potential (WP)’* of the thermal reservoirs” since it depends on their temperatures only, hence, being the logical proof of the claim presented here.
***KEYNOTE* 3**: Carnot heat engine efficiency dependence on the temperatures of the heat reservoirs would only be equally misplaced as if to attribute the maximum efficiency of an ideal waterwheel (water turbine), based on its motive power per unit of input water flow, and then it would also mistakenly depend on the water reservoirs’ elevations only. Therefore, all reversible devices are and have to be equally (not below nor above) the maximum possible 100% efficiency.

A motive power efficiency (i.e., a device’s *work efficiency*) should be consistently based on the *work potential* of an energy source (not on a “convenient nor arbitrary input quantity,” like the heat input or water flow input, etc.), and then, the ‘true’ *Carnot cycle efficiency* would be 100%, as for all the other efficiencies for ideal, reversible engines and motors.

Sadi Carnot defined engine cycle efficiency, logically and “empirically,” as “[work] motive-power *output per* heat *input*”, long before the concepts of the “*work potential*” of an energy source and *energy conservation* were established. However, we now have the advantage of looking at the historical developments more comprehensively and objectively than the pioneers [5,8,9,10,11], to put in order historical misconceptions.

An exact “*reverse*” of the reversible “*Power Carnot-cycle*” is the ideal “*Heat-pump cycle*” (“*Reverse Carnot-cycle*”), whose efficiency or “performance” is defined ‘in inverse’ as “heat *output per* work *input*”. It is always over 100% (as the “fundamental inverse” of the *Carnot cycle efficiency,* with the latter, as defined, always smaller than 100%), and it is named as the “*Coefficient of Performance (COP)*” since ‘such efficiency’ over 100% would not be fundamentally (nor “politically”) proper, but it should be fundamentally the same, i.e., improper also if smaller than 100%.
***KEYNOTE* 4:** For the same fundamental reason, the efficiency of a perfect, *ideal* Carnot cycle (being below 100%) would also be logically inappropriate (as if there are “some *work losses*” in the ideal reversible cycles). For the same reason as for the *Heat pump cycle*, it should be called the ‘*Carnot cycle COP*’, *not the ‘Cycle efficiency’*. Fundamentally, all ideal, reversible cycles must be “*equally and maximally [100%] efficient*,” as reasoned by Sadi Carnot [1].

Furthermore, it is fundamentally inappropriate, as often stated, to call the heat transferred out of the Carnot cycle at a lower temperature, the “*waste heat* or *loss*”, since it is the “*useful quantity,*” *necessary* for the completion of the perfect, ideal cycle. Together with the cycle work, they are reversibly transformable (interchangeable as equal) and present the “*reversible equivalent*” to the heat input at the high temperature, named by this author as the *Carnot–Clausius Heat Work Equivalency* (*CCHWE*) [11]), a crucially fundamental and autonomous concept (independent of any process or device), see also Figure 5. The only “waste or loss” that could lower efficiency below 100% would be any ‘*irreversible work dissipation’* (converted into generated heat) and accompanied by *entropy generation*, which must also be taken out to complete a real cycle. A device’s efficiency could not be higher than 100%, and it could only be lower due to irreversible, dissipative losses.

The “original,”, nowadays well-known *Carnot cycle efficiency* is misplaced and inappropriate, and it should be renamed for what it is: the *Work potential (WP*, o*r ‘available energy’) efficiency* of a heat source and sink, or *Exergy efficiency* of a thermal energy source with respect to the heat sink reference. We now know that the “true” Carnot efficiency, the *Second Law* or *Exergy efficiency*, is 100%. It is a goal here to clarify and rectify what is fundamentally misplaced. However, it would be hard “*to let go*” of the 200-year-long “habit and addiction”.

***Key NOVEL-Point 5*:*** The Carnot–Clausius [Ratio] Equality (CCE) and Clausius Equality (Cyclic integral) are special cases of a related “Entropy boundary integral” for reversible stationary processes*.

The balance equations (used for the definition of a new property, the *entropy*) were first developed by Clausius, based on Carnot’s discovery of “maximum efficiency and equality for all reversible cycles”, named here the *Carnot–Clausius Equality*, *CCE*, as the ratio *Q_H_/T_H_* = *Q_L_/T_L_* for the constant high and low temperature of the thermal reservoirs, to be the precursor of *Clausius Equality*, as a circular integral for a reversible cycle with variable temperatures, ∲δQ/T=0. Then, from those correlations, a new property, *entropy*, was inferred by Clausius, to be later generalized with *Clausius Inequality* as the *entropy balance*, as a “quantification” of the *Second Law* of thermodynamics.

The *Carnot–Clausius Equality* (*CCE,* as finalized and renamed here to reconcile and streamline it with the cyclic *Clausius Equality*, was named the *Carnot [Ratio] Equality*, *CtEq*, in [11]) is, in essence, the entropy balance, i.e., “*entropy-in* equal to *entropy-out*” of the reversible Carnot cycle at constant in and out temperatures, while the *Clausius Equality* is also the balance of the net entropy (*in-minus-out*) of a reversible cycle with varying temperatures, a cyclic integral around the cycle boundary or per cycle time period. They both represent special cases of the *entropy balance* for the steady-state, stationary processes (including quasi-stationary cyclic processes), where there is no accumulation of entropy (nor accumulation of any other system properties).

Note that engines are designed to run and produce power perpetually (except for necessary maintenance and repair). Therefore, their processes have to be either steady-state (stationary processes) or quasi-steady cyclic processes, often achieved by rotating or reciprocating piston and cylinder machinery, or any similar energy conversion devices. Neither steady-state nor cyclic processes accumulate mass and energy, but convert input to output while interacting with the energy reservoirs, an energy source and reference sink, the latter usually a surrounding device.

***Key NOVEL-Point 6*:*** The ‘caloric’ is transformable to work and cannot be ‘extended and renamed as entropy’, which is ‘the final transformation’*.

It is stated in *Miss Point 4* that “Sadi Carnot could not have been thinking of ‘any ‘other caloric’ but heat to imply the ‘entropy-like quantity’, as speculated by some”. There are creative and persuasive publications, albeit with fundamental deficiencies, that try to draw attention to and establish new interpretations of the caloric concept, first to ‘cancel’ its original meaning as “heat” (e.g., “Heat is not a noun” by Romer), and more recently as “Extended Caloric Theory” and to “make caloric equivalent to entropy”, as well as to generalize Carnot’s ‘waterfall analogy’ as the ‘Archetype of Waterfalls’ [16].

However, some critical statements are inconsistent with thermodynamic fundamentals. Namely, “*So, ‘any restoration of equilibrium in the caloric’ simply means any fall of caloric through a temperature difference*” [16]”. This is elusive and, in general, misplaced, since Carnot referenced the “restoration of equilibrium of caloric” via the reversible extraction of the motive power, as in an ideal heat engine, and not “any fall of caloric” like in an irreversible heat transfer from a high to low temperature in heat exchangers. Sadi Carnot also stated “*the need for reestablishing temperature equilibrium* [using adiabatic compression or expansion] *for* [isothermal/reversible] *caloric transfer*” (*Key Point II*). Furthermore, “*The authors’ motivation has been to make clear that an Extended Caloric Theory that allows for production of caloric, makes caloric equivalent to entropy in macroscopic thermodynamics* [16]”. This is misplaced and fundamentally incorrect. For example, the *caloric* may in part be isentropically converted to work and reduced (as in Carnot heat engine; entropy conserved), or dissipated to a lower temperature and conserved (as in a heat exchanger, while entropy will not be conserved); hence, the two are not the same (see also next and elsewhere).

Therefore, the *caloric*/heat is not equivalent to *entropy*; the latter is irreversible and the ‘*final transformation*’ (it is not possible to convert entropy to anything else—not possible to ‘destroy’ entropy), while the dissipative heat generation and/or conversion of heat to/from work, is not the ‘final transformation’ (caloric/heat could be increased or decreased, via a heat pump or heat engine, respectively, or generated by work dissipation). Furthermore, entropy is related to caloric (heat); the latter is the conjugate product of entropy (extensity) and temperature (intensity) [11]. It would be analogous and incorrect to claim that in a *waterwheel* the ‘water flow work’ is equivalent to the ‘water volume flow’, since ‘work’ is the conjugate product of ‘volume (extensity)’ and ‘pressure (intensity)’. Again, the caloric may be in part converted to work and vice versa (in “*thermal transformers*” i.e., heat pumps and heat engines) and thus overall increased or reduced, but the latter is not possible for entropy. Furthermore, *heat* (caloric) and *work* are reversibly transformable (interchangeable as equivalent), as formalized as the “*Carnot-Clausius Heat-Work Equivalency (CCHWE)*”, i.e., ‘heat at high temperature’ is (reversibly) equivalent, i.e., transformable to ‘work plus heat at low temperature’ and vice versa. As already stated, the *CCHWE* is a fundamental and autonomous physical concept, independent of any process or device [11], see also Figure 5.

Therefore, the *caloric* (thermal energy) is not *entropy* (thermal displacement/space). Certainly, conserved ‘*caloric*’ at Carnot’s time was known as the present name ‘*heat’,* which is extensively used in *calorimetry* nowadays, with “*calorie*” still being used as a heat unit, especially in chemistry and life.

## 5. Conclusions

The most logical and probable sequential developments by Sadi Carnot, regarding the *maximum efficiency of reversible cycles* and his ingenious reasoning of the *Reversible Contradiction Impossibility,* as well as the related consequences developed by his followers, and this author’s novel contributions and generalizations, are presented above and summarized in the *Key Takeaways* below.


**
*Key Takeaways:*
**
The conservation of *caloric* misconception was probably a fortunate catalyst leading to analogy with the *waterwheel* and Carnot’s hypothesis that the *motive power* of steam engine was caused and produced by the “*fall of caloric*” (reversible cooling of hot caloric), since Carnot believed that there was no “consumption” of the caloric (*Key Point I*);Carnot’s reasoning that “*wherever there exists a difference of temperature, motive power can be produced*” and not be wasted for ‘workless’ heat transfer, was the most critical and ingenious reasoning (*Key Point II*) that led to the inference of the most critical concept of Carnot’s discovery (*Key Points IV and V*);The “*Key discovery*” ingeniously inferred by Sadi Carnot that *Reversible cycles must all have equal and maximum efficiency*, by demonstrating that otherwise, would result in the creation of conserved caloric and/or perpetual motion: “*the maximum of motive power resulting from the employment of steam is also the maximum of motive power realizable by any means whatever* [1] (*Key Point V*)”*;*The selected and persistent post-misconceptions and fallacies by others are also presented as *Miss Points 1–4,* including the misconceptions that the heat transferred out of the ideal Carnot cycle at a lower temperature is the “*waste heat* or *loss*”, as often stated. However, it is the transformable component along with the motive power (i.e., a part of the heat–work interchangeability and ‘*Reversible Equivalency*’) “*useful and necessary quantity*” required for the entropy balance and completion of a perfect, reversible cycle, see Figure 4, Figure 5, Figure 6 and Figure 7;The “*Reversible and Reverse*” processes and cycles are re-examined in *Key NOVEL-Point 1*. Among others, it is emphasized that the potential qualities of flux quantities (heat and different kinds of works) are not degraded but equipotentially transferred and stored between the system and its boundary surroundings, and thus conserved in every way. The spatial gradients are virtually zero at any instant, while the time gradients and related fluxes may be arbitrary; hence, the *time and energy rates are irrelevant for the reversible analysis* of energy balances and properties;The reversible cycles “*Maximum efficiency*” and “*Reversible Equivalency*” are scrutinized in *Key NOVEL-Point 2*. The reversible efficiency implies the *maximum* work extraction and *minimum* work expenditure in the *reverse* process; therefore, the two must be equivalent for the given conditions, thus establishing the *‘Reversible Equivalency’.* The reversible processes are perfect and “*equally and maximally efficient,*”, and they define the concept of “*Reversible Equivalency*”—the *‘true equality’* of input and output, where relevant quantities and qualities are conserved in perpetuity;*The Reversible Contradiction Impossibility (*“*Reductio ad absurdum*”*)* is scrutinized in *Key NOVEL-Point 3.* Namely, any reversible cycle “*under-achievement*” (obtaining less than maximally possible) would become an “*over-achievement*” when such cycle is reversibly ‘*reversed’* (accomplishing with less than minimally required), and such a “*‘reversed’ over-achievement*” would be physically impossible (would be a “*‘Reversible Equivalency’ violation*” and may violate the *conservation laws*), thus establishing the “*Reversible Contradiction Impossibility*” of the established fact with numerous consequences, as detailed in *Key NOVEL-Point* 3 and elsewhere;Sadi Carnot [1] and his followers, including Kelvin [6] and Clausius [7], ironically referred to the maximum heat engine cycle efficiency they developed (at the time when most thermal concepts were unknown) with the *absurd conclusion* that it does not depend on the cycle design itself nor its operation mode; therefore, proving that *it is not the efficiency of ideal Carnot cycle ‘per se’*. Therefore, their attribution is misplaced, since the correlation they developed should have referred to the “maximum motive power or *‘work potential’* of the thermal reservoirs” since it depends on their temperatures only, hence being the proof of the claim presented here, see *Key NOVEL-Point* 4 and Figure 6;*The Carnot–Clausius [Ratio] Equality* and *Clausius Equality [Cyclic integral]* are elucidated to be the special cases of the related “*Entropy [balance] boundary integral*” *for reversible stationary or cyclic processes,* see *Key NOVEL-Point 5;*Finaly, in *Key NOVEL-Point* 6, it is reasoned why Sadi Carnot could not have been thinking of “any ‘other’ caloric” but heat to imply the “entropy-like quantity”, as speculated by some (see also *Miss Point 4*). Furthermore, the *caloric* (thermal energy) could not be *entropy* (thermal displacement/space) by any stretch of the imagination, as stated by some (e.g., [16]). It would be as erroneous as if the water energy in a waterwheel were claimed to be the water flow displacement. Certainly, conserved ‘*caloric*’ at the time was known as the present name ‘*heat’,* and was used as such by Carnot.


In conclusion, even though Sadi Carnot has often been named the “*Father of thermodynamics,*” with all fairness if conceivable, it might be more appropriate for Clausius and Kelvin to be named the *Fathers of thermodynamics*, since they meticulously developed the most critical concepts of thermodynamics, starting from thermodynamic temperature to entropy and the formulation of the *Laws* of thermodynamics, among others, whereas Sadi Carnot would be the “*Forefather of thermodynamics-to-become*” in honor of his ingenious discovery and reasoning of *heat engine reversible cycles and their maximum efficiencies* at a time when steam engines were in their initial development stage, the concepts of heat and work were not fully recognized, and even energy conservation was not established.

## Figures and Tables

**Figure 1 entropy-27-00502-f001:**
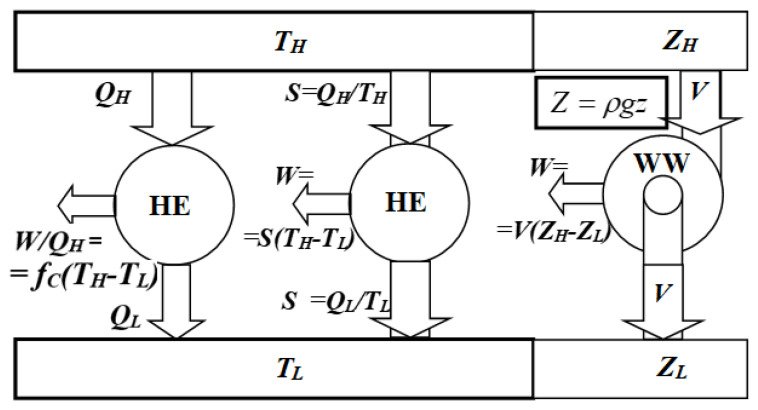
*Similarity between a heat engine* (HE) *and a waterwheel* (WW) [9]. At Carnot’s time, the ‘*caloric’* (heat) was considered to be conserved (*Q_H_* = *Q_L_* = *Q* and due to very low heat engine efficiency at that time, about 3%, or *Q_L_*≈0.97*Q_H_*≈*Q_H_,* so it was a reasonable misconception), leading to the *Waterwheel analogy*, with caloric flow (*Q_IN_* = *Q_H_*) and high (*H*) and low (*L*) temperature differences—although not linearly (**Left**), see Equation (1), being the proof that Carnot was not and could not have been thinking even remotely of any “*other’ caloric*” but heat, to imply “entropy-like quantity” as speculated by some, see *Miss Point 4*—corresponding to the water flow and elevation difference, respectively. It turned out, after discovery of entropy (*S* = *Q/T* = *constant* for reversible cycles [7]), that HE full similarity—with entropy and direct, linear temperature difference (**Center**)—with WW (**Right**) was established.

**Figure 2 entropy-27-00502-f002:**
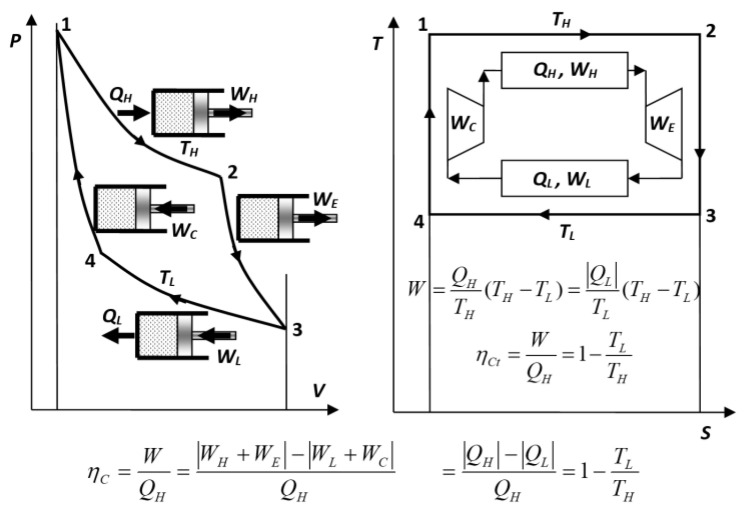
Heat engine ideal gas Carnot cycle [9]: note isothermal and adiabatic (mechanical) expansions (processes 1–2 and 2–3, respectively), and isothermal and adiabatic compressions (processes 3–4 and 4–1, respectively). The cycle net work out is realized via isothermal processes (*W* = *W_H_ − W_L_*), while adiabatic processes are needed to adjust temperatures with heat reservoirs to provide reversible, isothermal heat transfer, although their works cancel out (*W_E_* − *W_C_* = 0 [10]).

**Figure 3 entropy-27-00502-f003:**
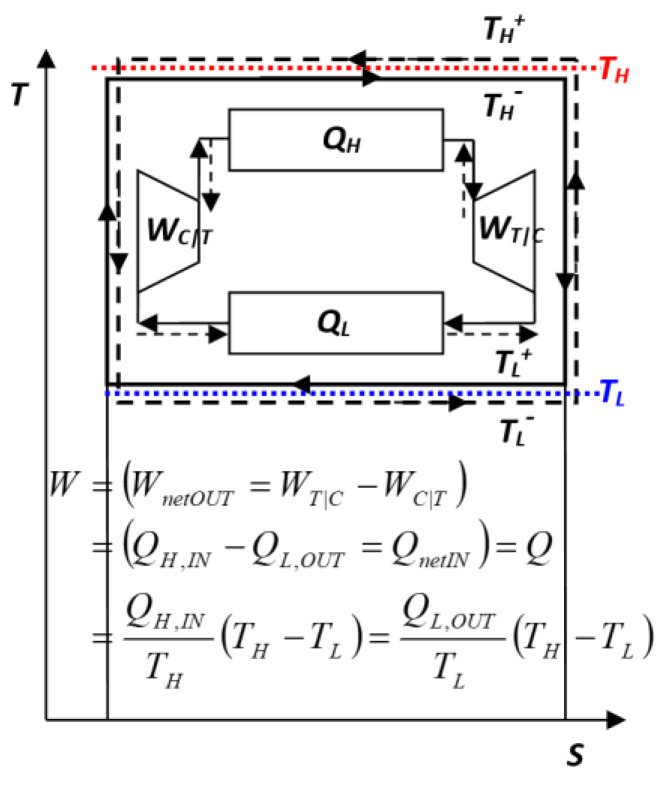
Reversible steam engine Carnot power cycle (*solid lines*) and ‘reversed’ Carnot refrigeration cycle (*dashed lines*, reversed directions) [9]. Note: infinitesimal temperature differences above (*T^+^*) or below (*T^−^*) thermal reservoir temperatures, *T_H_* (in *red*) and *T_L_* (in *blue*), to provide heating or cooling as needed.

**Figure 4 entropy-27-00502-f004:**
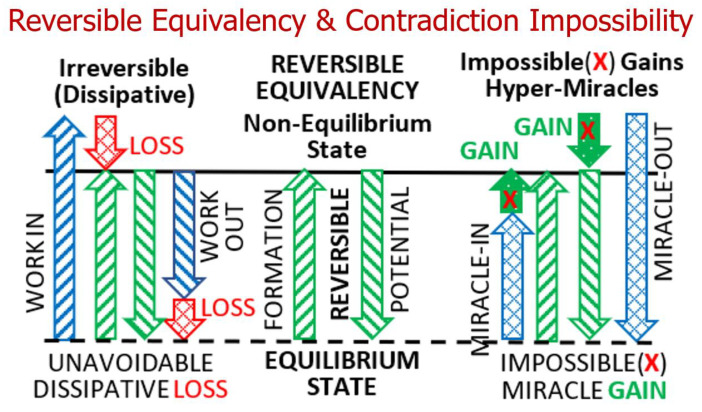
*Reversible Equivalency and Contradiction Impossibility*: Ideal *formation work* of “non-equilibrium state” (minimum required work with perfect efficiency) is equal to its *work potential,* WP (maximum available work with 100% perfect efficiency (no loss; **Center**)), otherwise it will result in *Contradiction impossibility *(**Right**). Formation of non-equilibrium state with less than its WP or obtaining more useful work than WP would require a “*miracle work-GAIN*” without due work source (violation of *Second Law*), being against natural forcing and existence of equilibrium, thus being impossible (**Right**). However, real formation *work in* is bigger, and retrieved useful *work out* is smaller than its stored WP (**Left**) due to unavoidable dissipation loss, resulting in lower than 100% perfect efficiency (**Left**). Furthermore, any ideal, reversible ‘underachievement’ when reversed would become ‘overachievement’ and form WP with less work, resulting in impossible, miracle gain, as well as ‘overachievement’ (over WP; **Right**), i.e., all reversible processes must be equally perfect with 100% efficiency (**Center**).

**Figure 5 entropy-27-00502-f005:**
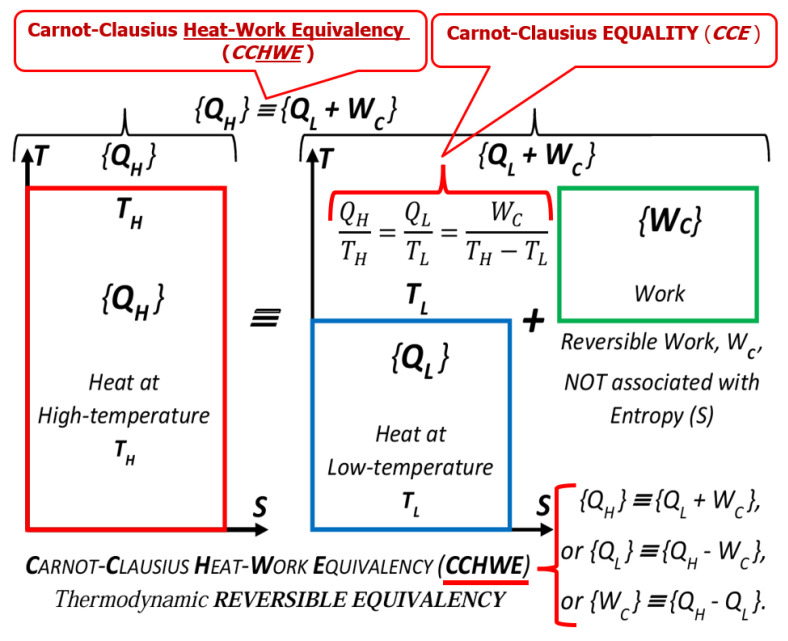
The *heat* may in part be converted to *work*, and vice versa (in “*thermal transformers*” i.e., heat engines and heat pumps), i.e., heat and work are reversibly transformable (interchangeable as equivalent), generalized, and named as the “*Carnot-Clausius Heat-Work Equivalency (CCHWE)*”, as “*‘Heat at high temperature’ is (reversibly) equivalent,* i.e.*, transformable, to ‘work plus heat at low temperature’*”. The *CCHWE* is a fundamental and autonomous physical concept, independent of any process or device [11]. The *Q/T* = *constant* is an invaluable but not well-recognized correlation, named here “*Carnot-Clausius Equality (CCE)*” as the precursor to and to resemble the ‘*Clausius Equality*’ cycle integral, both formalized by Clausius [7] based on Carnot’s *Réflexions* [1], see Equation (4).

**Figure 6 entropy-27-00502-f006:**
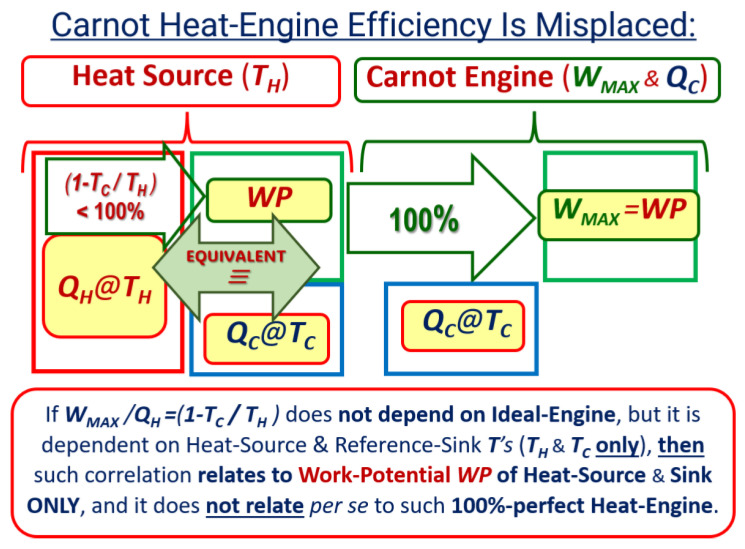
*Carnot heat engine efficiency is misplaced*: The reversible processes and cycles (power and reverse), as a matter of concept, are 100% perfect without any degradation and must be equally and perfectly (maximally) efficient, not over nor below 100% (‘*Reversible Equivalency*’, see Figure 4). Carnot defined engine efficiency, logically and “empirically,” as “[work] motive-power output per heat input,” thus implying both the engine and heat reservoirs’ combined system. Since ideal engines are maximally 100% efficient (**Right**) and overall efficiency depends on the reservoirs’ temperatures only, not on the engine in any way, then famous (1 − *T_C_*/*T_H_*) Carnot efficiency refers to the former only (**Left**). However, it would be hard “to let go” of the 200-year-long “habit and addiction”.

**Figure 7 entropy-27-00502-f007:**
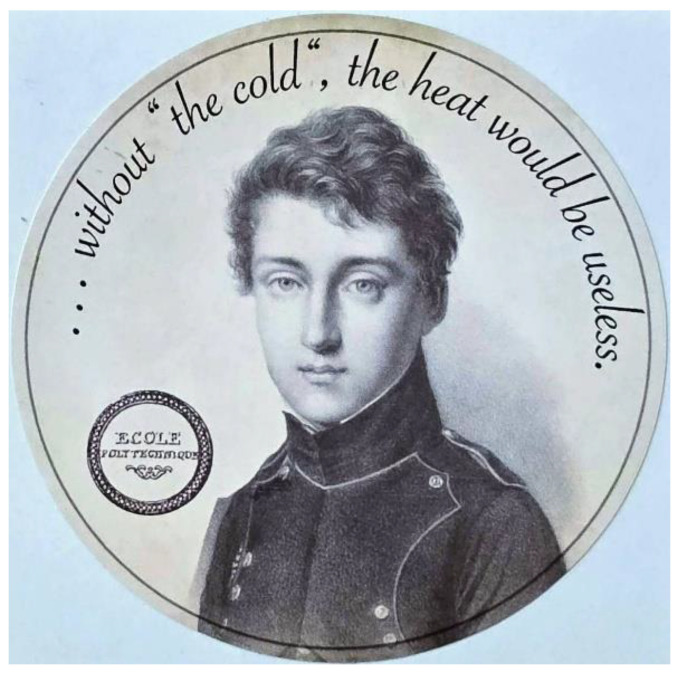
*Carnot Legacy STICKER: “…**without ‘the cold’, the heat would be useless**”*, École Polytechnique [3]. It illustrates claims in *Key Point I* and *Miss Point 3: “The [cold] heat transferred out of the Carnot cycle at lower temperature is “not a waste heat” as often stated, but it is “useful quantity”, necessary for completion of the cycle”. “The cold”* or heat at low temperature is actually a component, in addition to work, of the heat at high temperature. Namely, heat and work are reversibly transformable (interchangeable as equivalent), generalized, and named as *“Carnot-Clausius Heat-Work Equivalency (CCHWE)”* as *“‘Heat at high temperature’ is (reversibly) equivalent, i.e., transformable, to ‘work plus heat at low temperature’”*. The CCHWE is a fundamental and autonomous physical concept, independent of any process or device [11], see also Figure 5.

## Data Availability

The original contributions presented in this study are included in the article. Further inquiries can be directed to the author.

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
