# Peer review of "2024 ‘Key Reflections’ on Sadi Carnot’s 1824 ‘Réflexions’ and 200 Year Legacy"

_entropy, 2025, doi:10.3390/e27050502_

Round 1
Reviewer 1 Report
Comments and Suggestions for Authors
Review and recommendation
Early Comments to the Author
- The author presents ''Key Reflections on the 1824 Sadi Carnot’s ‘Réflexions'' and in regard with ''200-Year ‘Réflexions’ Anniversary''. (p. 1). But, from the beginning (Abstract) he declares he ''[…] is not a philosopher nor historian of science, but an engineering thermodynamicist. In that regard and in addition to various philosophical “why & how” treatises and existing historical analyses, the physical and logical “what it is” (p. 1) reflections, as successive Key Points, where a key reasoning infers the next one, along with novel contributions and original generalizations, are presented''. (p. 1). In this sense the aim and related content of the article is confused, without scientific method nor historical one.
- Several scientific and historical assumptions are uncertain/imprecise and a historical-scientific confusion arises. Thus, as well as the Conclusion.
- The author should really improve–rework presentations, included its lay-out, as well as for making sure it best conveys the author's ideas.
Main Specific Comments to the Author
- ''Introduction''
Certain historical claiming (for example on Carnot's death, caloric, etc.) are historically obsolete with respect to current literature and correlated evidences on the subject.
Certain scientific assumption should be clarified such as not belonging to Sadi Carnot own book. From historical standpoint method, it is very important to declare when and where an ancient scholar claims and when and where w amodern scholar modernize the wording. Otherwise, the result is a confusion as above announced.
- ''Key Points'', ''Miss Points'' and ''NOVEL-Points''
Certain scientific-historical claiming (for example ''[…] in par with Einstein's Relativity theory in modern terms […]'', p. 6 and Eq. (4) ) should seriously be rethought.
''Hot'' and ''cold'' (p. 7). If the author refers to them as historical terms, then the he should include the pages of the sources, in order to give the possibility to the reader who are not familiarity with Sadi Carnot's book to find the claiming (if Sadi Carnot exactly and really wrote that). This is valid for all other italic style quotations claimed without citing the number of pages. From historical standpoint, this way to write is unacceptable. But the author, from the beginning, declared that he ''[…] is not a philosopher nor historian of science, but an engineering thermodynamicist'' (p. 1), so again a methodological confusion, within the article, arises.
An example: ''Ideal and perfect, “Reversible processes” take place at infinitesimal potential difference (temperature, pressure and similar) at any instant within and between a system and its boundary surroundings, but they may and do change in time (process is a change in time). Namely, the spatial gradients are virtually zero at any instant while time gradients and related fluxes may be arbitrary as driven by ideal boundary surroundings and facilitated by ideal arbitrary (or infinite) transport coefficients. Therefore, the potential qualities of flux quantities (heat and different kinds of works) are not degraded but equipotentially transferred and stored between the system and its boundary surroundings, and thereby ‘truly’ conserved in every way. However, in time, due to unavoidable irreversible dissipation of work to generated-heat, accompanied by generation of entropy, all real processes between interacting systems (including relevant surroundings) are asymptotically approaching common equilibrium with zero mutual work potential and maximum mutual entropy.'' (p. 10). Did Sadi Carnot write or discussed it? Not at all: other methodological confusion arises.
Etc.
- ''Conclusion and “Key Takeaways” ''
The rewriting author's key points (including "'Novel-Points'') related to Sadi Carnot Thermodynamics are already discussed, defined and investigated (by means of scientific-logical and historical methods) in the literature.
Finally this is seems to be the aim of the article
''In conclusion, even though Sadi Carnot has been often called as the “Father of thermodynamics,” with all farness if conceivable, it might be more appropriate for Clausius and Kelvin to be named as the Fathers of thermodynamics […] whereas Sadi Carnot would be the “Forefather of thermodynamics-to-become” […]'' (p. 19)
References
The references on the subject are not-exhaustive. The risk is to have produced too many auto-referred isolates assumptions.
Several current and advanced historical literatures on the subject lack in the author's paper. Idem for advanced historical-scientific research literatures. I suggest reading some recent literature on the subject, in order to discover the amount of studies listed. Of course, it is not thinkable to add many references in a paper, but the main are necessary to list.
In addition, more recent (and better) transcriptions and translations of Sadi Carnot's book are suggested (see: Robert Fox 1978 and 1986).
Final Recommendations to the Author
- The tentative of the author to discuss historical claiming of Sadi Carnot (without quoting the pages) from modern standpoint produced misconceptions from both historical standpoint and scientific one.
- At this stage the paper is too weak and so not ready for a detailed academic publication on the subject.
- The manuscript should be re-considered further.
- The manuscript is rejected.
- The author is friendly advised.
Author Response
Please see attached PDF file.

Reviewer 2 Report
Comments and Suggestions for Authors
It is not obvious to this referee that the paper contains new insights that go beyond what the author has already written in his previous publications [9-11]. The paper is also not well written. Section 4, with the title "Key NOVEL-points: Novel contributions and original generalizations" is quite verbose but the statements are either rather trivial or ambiguously formulated. Example of the former: The discussion of `reversible´ vs `reverse´ in "Key NOVEL-point 1"; example of the latter: `Reversible contradiction impossibility´. Also, "Key NOVEL-point 5" is well known and hardly qualifies as a `novel contribution´.
Comments on the Quality of English LanguageFormulations in the text are rather clumsy in places. An example is the second sentence of the Abstract: "In that regard and in addition to various philosophical “why & how” treatises and existing historical analyses, the physical and logical "what it is" reflections, as successive "Key Points", where a key reasoning infers the next one, along with novel contributions and original generalizations, are presented."
Author Response
Please see attached PDF file.

Reviewer 3 Report
Comments and Suggestions for Authors
The manuscript meticulously reviews Sadi Carnot's fundamental philosophical contributions that subsequently lead to the formulation of classical thermodynamics, addressing common misunderstandings about his concepts and emphasizing their enduring importance. The presentation is clear and well-structured, and effectively clarifies key points and concepts, including mentioning dissipation and entropy production, which are often misunderstood in scientific and engineering disciplines. I found the manuscript very interesting and strongly recommend it for publication.
Author Response
Please see attached PDF file.

Reviewer 4 Report
Comments and Suggestions for Authors
See attached file.

Author Response
See the attached file.
Comment to the Editor:
This Author, as retired, lifelong engineering-thermodynamicist and professor emeritus of mechanical engineering at Northern Illinois University,[1] a Professional Engineer (PE) in Illinois, and Section First Editor-in-Chief of Thermodynamics (2015-2024) of the journal Entropy[2], like to respectfully and sincerely respond as follows:
I respectfully appreciate the reviewers’ comments and suggestions, although not agreeing with most of them, as detailed in my “Responses to the Academic Editor and [prior] Reviewers” (in a separate PDF file), and below.
I consider that the review process was prejudiced and unprofessional (inadequate and misleading). It was missing the main thermodynamic aspects of the manuscript, as I have stated in my Point-to-point Responses [3] that have been “unnoticed or intentionally ignored,” see also below.
It appears, considering the substance of the Reviewers’ comments, that the Academic Editor selected the Reviewers who are not specialists in the ‘phenomenological thermodynamics’ (that is more general and subtle to comprehend than the other special areas of thermodynamics), who, therefore, could not comprehend the subtle and critical contributions presented. Sometimes, highly accomplished scientists in their fields do not fully comprehend the essence of thermodynamics, especially if related to the Second Law and Entropy, as demonstrated by the Reviewers’ comments.
Regrettably, the additional Reviewer #4 (R4), has not resolved the disputed issues by his question able comments that are conflicting with the well-established knowledge of classical, phenomenological thermodynamics. Namely, R4 stated among others, “Reversible arbitrary cycles have not maximum efficiency, which can depend on the working substance, e.g. Stirling’s,” [as if “… not all reversible cycles are equally efficient … as if the ideal Stirling cycle is less efficient than the Carnot cycle, etc.],” and by repeated confusing ‘proclamation’, “historical and personal interest” [as if irrelevant but without due justification], as if to avoid and “cancel’’ important thermodynamic aspects presented, see Author’s responses to all R4’s Line #s items, see attached PDF file.

Round 2
Reviewer 1 Report
Comments and Suggestions for Authors
The revision is not sufficient and appropriate to publish the article.
Author Response
See attached PDF file
